# 3D Model Retrieval Algorithm Based on DSP-SIFT Descriptor and Codebook Combination

**Yuefan Hu [1], Haoxuan Zhang [2], Jing Gao [3,\*] and Nan Li [2]**

[1] Computational Aerodynamics Institute, China Aerodynamics Research and Development Center, Mianyang 621000, China
[2] School of Computer Science and Engineering, Beijing Technology and Business University, Beijing 100048, China
[3] Information Network Center, Beijing Technology and Business University, Beijing 100048, China
\* Correspondence: gaojing@btbu.edu.cn

**Abstract:** Recently, extensive research efforts have been dedicated to view-based 3D object retrieval, owing to its advantage of using a set of 2D images to represent 3D objects. Some existing image processing technologies can be employed. In this paper, we adopt Bag-of-Words for view-based 3D object retrieval. Instead of SIFT, DSP-SIFT is extracted from all images as object features. Moreover, two codebooks of the same size are generated by approximate k-means. Then, we combine two codebooks to correct the quantization artifacts and improve recall. Bayes merging is applied to address the codebook correlation (overlapping among different vocabularies) and to provide the benefit of high recall. Moreover, Approximate Nearest Neighbor (ANN) is used to quantization. Experimental results on ETH-80 datasets show that our method improves the performance significantly compared with the state-of-the-art approaches.

**Keywords:** view-based 3D model retrieval; Bag-of-Words; codebook combination; Bayes merging

## 1. Introduction

With the rapid development of computer science, 3D models have been widely used for applications such as 3D movies, 3D graphics, CAD, 3D architectural design, etc. Due to the explosive growth of the number of 3D models in recent years, how to accurately find the desired 3D model among the massive number of 3D models and improve the 3D model's reuse rate has become an urgent problem to be solved [1]. There have been some developments in 3D model retrieval [2]. Deep learning has been introduced to 3D reconstruction [3], such as the method based on generative adversarial networks [4]. Meanwhile, deep learning also has a lot of development space in 3D data analysis and understanding [5].

In general, the purpose of content-based 3D model retrieval is to find a 3D model which is similar to the input at the content level. Usually, content-based 3D model retrieval can be divided into the following steps: (1) Input the model to be retrieved; (2) Extract the feature descriptor of the model to be retrieved; (3) Define a suitable retrieval method to automatically calculate the similarity distance between the models; (4) Output search results according to similarity distance ranking [6].

According to different types of 3D model data, the existing content-based 3D model retrieval algorithms [7–9] can be roughly divided into two categories: (1) Model-based 3D model retrieval methods; (2) View-based 3D model retrieval methods. In model-based 3D model retrieval methods, model geometry information [10], surface area distribution [11], volume information [12], surface area geometry information [13], and so on are usually used to describe the 3D model. For most existing model-based 3D model retrieval methods, it is difficult to obtain model information of objects in practical applications. If the

model does not exist, this type of method may need to generate the desired three-dimensional model through the image of the model. The process of 3D modeling takes a lot of time. Meanwhile, the choice of image will also affect the accuracy of the algorithm. It extremely limits the application of this type of method.

In the view-based 3D model retrieval method, each object is represented by multiple images from different angles. Figure 1 shows three different models' multiple views. These images can be obtained by a set of cameras or a sequence of virtual cameras. Therefore, the view-based 3D model retrieval can be transformed into a group matching problem between image sets and image sets. View-based 3D model retrieval has become a new research hotspot [14–16]. Compared with model-based 3D model retrieval methods [17–19], this type of method has the following advantages: (1) The view-based 3D model retrieval method is more flexible because it does not require virtual 3D model information; (2) The retrieval accuracy of rigid models and partial matching is relatively high; (3) The existing image processing technology can be used to improve the retrieval accuracy; (4) The input requirements are reduced, which is conducive to the use of sketches and 2D images as input for retrieval. Therefore, view-based 3D model retrieval is widely used.

Deep learning is a new field of machine learning. Deep learning has been widely used in 3D model retrieval and has achieved excellent performance. Moreover, 3D model retrieval methods based on deep learning can be divided into three research directions depending on their input modes, namely, voxel-based methods [20–25], point-set-based methods [26–30], and view-based methods. An object is represented as a 3D mesh in the voxel-based methods and is analyzed by a 3D network. In the point-set-based methods, an object is represented as a set of unordered points, and the point cloud is used for prediction. These two methods can also be collectively referred to as model-based methods. The model-based methods use a 3D convolution filter to convolute a 3D shape in 3D space, thus generating a 3D representation directly from the 3D data [31,32]. The view-based methods render 3D objects to 2D images from different viewpoints and convolute these views using a 2D convolution filter. The view-based methods do not rely on the complex 3D features, and it is easy to capture the input view in these methods. They have a large amount of data and can make use of a mature advanced network framework.

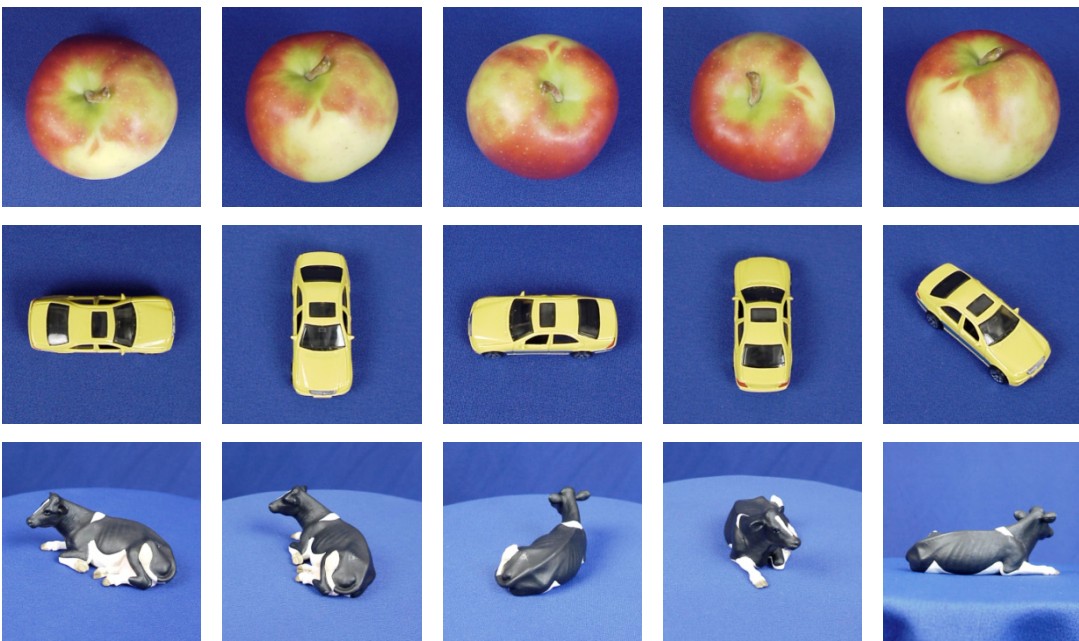

**Figure 1.** Some views of the model datasets on ETH-80 (one model per row).

This paper designs an improved Bag-of-Words model and applies it to view-based 3D model retrieval. First, we extract DSP-SIFT as a model feature, and then, we generate

two codebooks in the same size for combination. In order to solve the problem of code-book correlation, the Bayes merging algorithm [33] is introduced to reduce local quantization errors and improve recall. The experimental results show that our method improves the performance significantly comparing with the state-of-the-art methods. Our method has the following advantages:

- Applying the Bag-of-Words model to the 3D model retrieval. The existing image processing technology can be used, and good retrieval results have been obtained.
- Extracting DSP-SIFT features and improving the Bag-of-Words model in the feature extraction stage.
- Improving the Bag-of-Words model through codebook combination, which corrects quantization artifacts of local features. Additionally, the Bayes merging algorithm is used to address the codebook correlation and improve the accuracy of the algorithm.

## 2. Bag-of-Words

In V3DOR (View-based 3D Object Retrieval), the 3D model is represented by a set of images, and the Bag-of-Words model can be used in this field. Takahiko Furuya and Ryutarou Ohbuchi [34] first introduced the Bag-of-Words model to this field. In this method, the 3D model is represented by a set of depth images, and all image SIFT operators are extracted as model features. After the codebook generation, the feature histogram of model is generated, and the model similarity is calculated. This method simply uses the Bag-of-Words model for 3D model retrieval. Later, Ohbuchi et al. [35] improved this; they used the KL distance (Kullback-Leibler divergence) to calculate the similarity distance between models. In addition, Ohbuchi et al. [36] proposed an acceleration algorithm to further accelerate the above algorithm and improve retrieval efficiency. Gao et al. [37] improved the Bag-of-Words model and proposed a Bag-of-Region-Words model. This method divides the image into different regions, assigns different weights, and further extracts BoRW features and calculates the similarity between models. Experiments show that the retrieval effect of the original bag-of-words model is slightly improved. Alizadeh et al. [38] proposed a new feature descriptor and simply used the Bag-of-Words model to calculate the similarity distance between models.

Overall, the above methods are the application or improvement of Bag-of-Words in the view-based 3D model retrieval. Compared with the above algorithm, our method uses codebook combination to improve the Bag-of-Words model, and introduces the Bayes merging algorithm to further eliminate codebook cross-correlation and improve retrieval accuracy.

## 3. Proposed Method

This section introduces in detail the 3D model retrieval algorithm using the Bayes algorithm for codebook combination. First, we extract the DSP-SIFT features of the input model. After the feature extraction is completed, the approximate k-means algorithm is used to generate two codebooks with the same scale. For a given feature, it is quantified into a visual word in two codebooks. Finally, we introduce the Bayes merging algorithm to combine two codes, which reduce the quantization error and improve the retrieval results' recall. The flowchart of our method is shown in Figure 2.

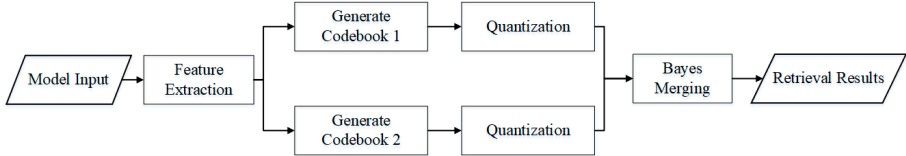

**Figure 2.** The flowchart of the algorithm.

### 3.1. Feature Extraction

Before feature extraction, the contour image will be used to remove the background of the RGB image. Different from other methods, our method extracts DSP-SIFT features instead of SIFT features. Because DSP-SIFT extracts key points from images with different sampling scales, it is more representative than SIFT features. The DSP-SIFT extraction process is shown in Equation (1):

$$h_{DSP} = (\theta|I)[x] = \int h_{SIFT}(\theta|I,\sigma)[x]\varepsilon_s(\sigma)d\sigma \quad x \in \Lambda \tag{1}$$

where $\theta$ is the direction of the key point ($0 < \theta < 2\pi$), $I$ refers to the squared image, $x$ is the image coordinates, $h_{SIFT}$ refers to the SIFT extraction method, $\sigma$ refers to the degree of Gaussian difference scale space, $S > 0$ is the scale of size-pooling, and $\varepsilon$ is an exponential or one-sided density function. The DSP-SIFT feature extraction steps are shown in Figure 3: (1) Zoom the image to find the key points; (2) Zoom the image to its original size, and take the multi-scale convolution kernel to convolve the key points; (3) Extract SIFT features with different scales; (4) Integrate all SIFT features and make a histogram; (5) Normalize the obtained descriptor to the same dimension as the SIFT operator.

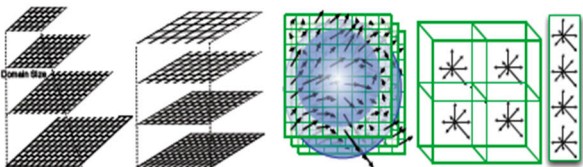

**Figure 3.** The DSP-SIFT feature extraction process.

### 3.2. Codebook Combination

After the feature extraction stage, the approximate k-means algorithm is used to generate two codebooks with the same scale. After the quantization of the ANN algorithm is completed, our method combines two codes to reduce the quantization error and improve the recall. The advantage of codebook combination is that more candidate features can be used, which reduces the error generated by the quantization process to a certain extent. Since our method uses the same feature to generate codebooks, the correlation between codebooks, that is, crossover between codebooks, is inevitable. As a result, when calculating the similarity distance, the features of the intersection will be repeatedly calculated, which reduces the retrieval accuracy. The codebook crossover problem is shown in Figure 4. For a given feature, it is quantified into a visual word in two codebooks. Then, the indexes of the two visual words $A$ and $B$ are respectively determined in the two index files. $A \cap B$ in Figure 4 represents the crossover problem between two sets.

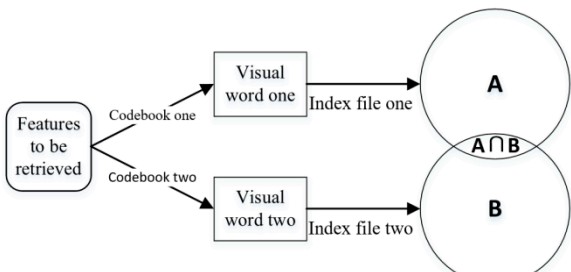

**Figure 4.** Codebook correlation problem.

In order to solve this problem, we introduce the Bayes merging algorithm. The algorithm is defined as follows: For a given $N$ codebooks, the feature $x$ of the model $Q$ to be retrieved is quantized into $N$ visual words, and the index of $N$ sets corresponding to

$x$ is also determined, such as $\{A_i\}_{i=1}^{N}$. If feature $y$ falls in $n^{th}$ intersections which are in all set $\{A_i\}_{i=1}^{N}$, then the conditional probability of $x$ and $y$ matching is defined as:

$$w(x, y) = p(y \in T_x | y \in A_1 \cap A_2 \cap \ldots \cap A_N) \tag{2}$$

where $T_x$ represents a feature set similar to feature $x$ in a model similar to the model $Q$ which is retrieved. Let $F_x$ be the inverse of $T_x$, then $T_x$ and $F_x$ satisfy the following formula:

$$p(y \in T_x) + p(y \in F_x) = 1 \tag{3}$$

substituting Bayes' rule into Equation (3), we can obtain:

$$p(T_x | A \cap B) = \frac{p(A \cap B | T_x) \cdot p(T_x)}{p(A \cap B)} = \frac{p(A \cap B | T_x) \cdot p(T_x)}{p(A \cap B | T_x) \cdot p(T_x) + p(A \cap B | F_x) \cdot p(F_x)} \tag{4}$$

where $A \cap B$ represent $y \in A \cap B$, $T_x$ represents $y \in T_x$, $F_x$ represents $y \in F_x$. Finishing Equation (4), the probability of intersection between codebooks after matching can be obtained:

$$p(T_x | A \cap B) = \left(1 + \frac{p(A \cap B | F_x)}{p(A \cap B | T_x))} \cdot \frac{p(F_x)}{p(T_x))}\right)^{-1} \tag{5}$$

in the final matching stage, the Bayes merging algorithm matching equation is defined as:

$$f(x, y) = \begin{cases} nw(x, y), & if \ y \in \cap^n, n \geq 2 \\ \sum_{i=1}^{n} \delta_{v_x^{(n)}, v_y^{(n)}} & otherwise \end{cases} \tag{6}$$

the steps of the Bayes merging algorithm are as follows: (1) Quantify the feature $x$ into $N$ visual words; (2) Determine the index of $N$ sets; (3) Find all intersections of $N$ sets; (4) Find all collections of $N$ sets; (5) For each feature in $N$, find all the intersections and collections where it is located, and calculate the ratio of the two sets, use Equation (5) to remove the intersection, use Equation (2) to find its matching feature, and use Equation (6) to vote and obtain matching images.

## 4. Experiments

### 4.1. ETH-80 Datasets

Our experiments use the ETH-80 [39] datasets. The ETH-80 dataset contains visual object images from eight different categories, including apples, cars, cows, cups, dogs, horses, pears, and tomatoes. For each category, there are 10 object instances and 41 images for each object instance captured from different viewpoints. Figure 1 shows a partial model of the ETH-80 datasets.

### 4.2. Evaluation Metrics

In this paper, we use SHREC competition [37] general evaluation metrics, as follows:

1. P-R curve: P-R curve evaluation metrics are widely used in information retrieval systems. The precision rate refers to the proportion of relevant results in the search results. Recall refers to the proportion of relevant search results in the entire datasets among the search results. Let $A$ represent all relevant results in the datasets and $B$ represent all search results, then:

$$precision = \frac{A \cap B}{B} \qquad recall = \frac{A \cap B}{A} \tag{7}$$

2. F-measure ($F$): $F$ is the weighted harmonic average of precision and recall, and is a commonly used retrieval metrics in information retrieval systems. $F$ can be defined as (taking the first 20 search results in the experiment):

$$F = 2 \times \frac{precision \times recall}{precision + recall} \tag{8}$$

3. $NN, FT, ST$: The evaluation methods of these three evaluation standards are similar. Searching the first $K$ test results, the proportion of the same category as the retrieved object is tested. Suppose there are $|C|$ objects in the category where the search object is located, and if $K = 1$, it is $NN$. $K = |C| - 1$ represents $FT$. $K = 2 \times (|C| - 1)$ represents $ST$. The final result of the three evaluation metrics is the average of the retrieval results of all objects in the datasets.

4. $DCG$: $DCG$ describes the location information of the relevant result in the search result. The higher the relevant result in the search ranking, the greater its weight. The value is between 0 and 1. The larger the value, the better the search result. The definition of $DCG$ is as follows:

$$DCG_1 = G_1; \ DCG_i = DCG_{i-1} + \frac{G}{\lg_2(i)}, \quad if \ i > 1 \tag{9}$$

5. The final result is defined as:

$$DCG = \frac{DCG_k}{1 + \sum_{j-1}^{|C|} \frac{1}{\lg_2(i)}} \tag{10}$$

*4.3. Qualitative Results*

4.3.1. Codebook Size

The size $k$ of the codebook, that is, the number $k$ of clustering centers of approximate k-means, may have a direct impact on the effect of the algorithm. We take different values of $k = 300, 500, 700, 900, 1100$ for comparison and determined the best. The results are shown in Figures 5 and 6. This paper takes $k = 1100$ as the final result of the algorithm.

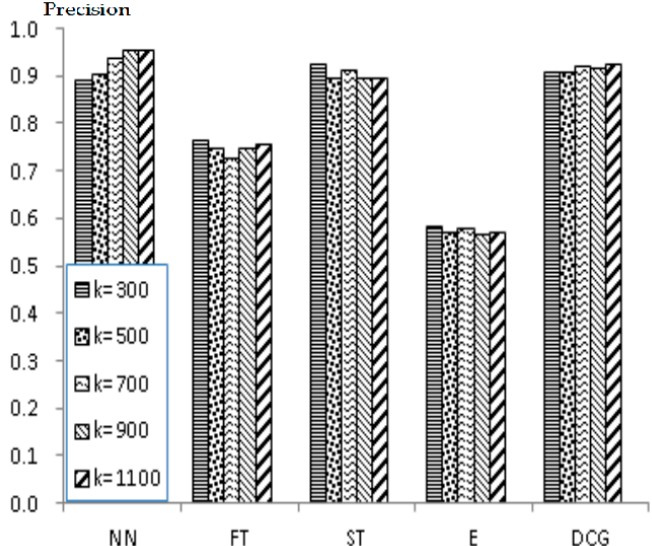

**Figure 5.** The five evaluation criteria of each codebook size $k$.

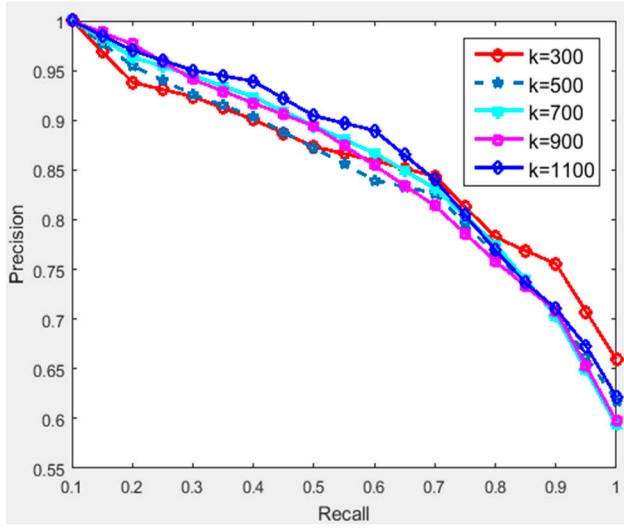

**Figure 6.** PR curve of different codebook size $k$.

### 4.3.2. The Effectiveness of Our Method

Different from other methods, we extract DSP-SIFT features instead of SIFT features as model features. Table 1 and Figure 7 show the comparison of retrieval effect between our method (Ours) and SIFT+Bayes when only the features are different. Then, we use the Bayes merging algorithm to eliminate cross-correlation after codebook combination. Table 1 and Figure 7 show the comparison between our method (Ours) and the single codebook retrieval algorithm (C1, C2) using DSP-SIFT features.

**Table 1.** The five evaluation criteria of each step of our method.

| Method | NN | FT | ST | F | DCG |
|---|---|---|---|---|---|
| C1 | 0.9500 | 0.7472 | 0.8917 | 0.5690 | 0.9170 |
| C2 | 0.9250 | 0.7514 | 0.8903 | 0.5638 | 0.9210 |
| SIFT+Bayes | 0.9250 | 0.6889 | 0.8694 | 0.5517 | 0.8860 |
| Ours | 0.9500 | 0.7528 | 0.8931 | 0.5698 | 0.9220 |

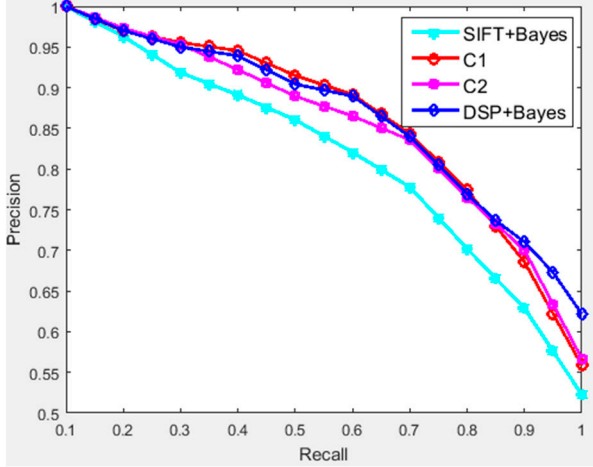

**Figure 7.** PR curve of each step of our method.

### 4.3.3. Comparison with Existing Methods

We compared our method with the existing algorithms, including MMGF, BoRW [37], BGM [40], AVC [41], CCFV [42], and FDDL [43]. From the experimental results, as shown in Figures 8 and 9, the following results can be obtained. It can be seen from Section

4.3.1 that, overall, the retrieval results of our method vary with the value of $k$, and the performance is relatively stable. This article takes $k = 1100$ as the final result. Section 4.3.2 can prove that each step of the algorithm improves the retrieval results to varying degrees. The algorithm result using DSP-SIFT is better than the algorithm result using SIFT. The algorithm result after codebook combination is better than the retrieval result of single codebook algorithm. Compared with the existing algorithms through this section, our method has better retrieval accuracy.

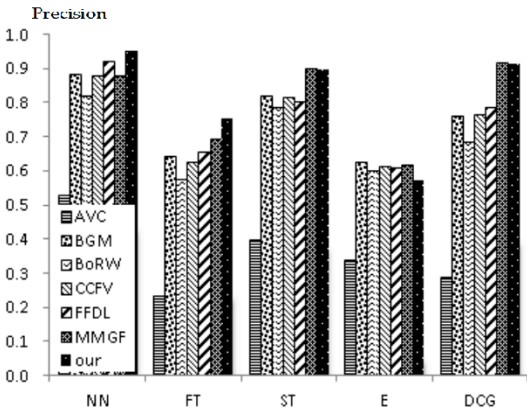

**Figure 8.** The five evaluation criteria of five different methods and ours.

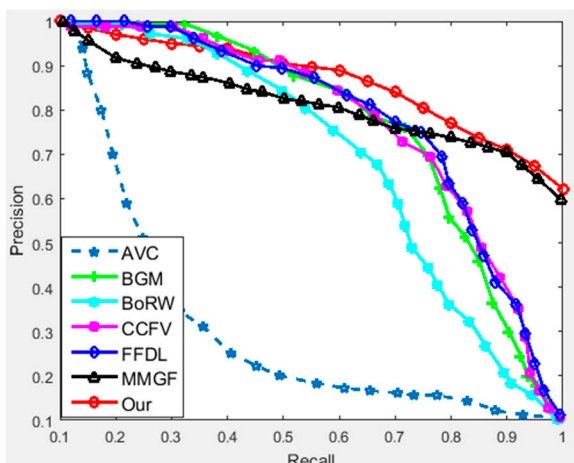

**Figure 9.** PR curve of five different methods and ours.

## 5. Conclusions

In this paper, the Bag-of-Words model is improved and applied to the view-based 3D model retrieval, and good retrieval results are obtained. Different from other methods, we extracts DSP-SIFT as a model feature, and uses Bayes merging algorithm for codebook combination to improve the retrieval effect. Experiments verify the effectiveness of each step of the algorithm. At the same time, because the algorithm does not require a virtual 3D model as input, the algorithm is more flexible in practical applications. Subsequent work can focus on the association of images, using view learning and other related methods to eliminate redundant information between images, and further improve the efficiency of the algorithm.

**Author Contributions:** Conceptualization, Y.H. and H.Z. Writing—original draft preparation, Y.H. Writing—review and editing, Y.H., H.Z., J.G., and N.L. All authors have read and agreed to the published version of the manuscript.

**Funding:** This work was partially supported by the Beijing Natural Science Foundation and Fengtai Rail Transit Frontier Research Joint Fund (grant No. L191009), the National Natural Science Foundation of China (grant No. 61877002, No. 62277001), and the Scientific Research Program of Beijing Municipal Education Commission (grant No. KZ202110011017).

**Institutional Review Board Statement:** Not applicable.

**Informed Consent Statement:** Not applicable.

**Data Availability Statement:** Not applicable.

**Conflicts of Interest:** The authors declare no conflict of interest.

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
