# Peer review of "3D Model Retrieval Algorithm Based on DSP-SIFT Descriptor and Codebook Combination"

_applsci, doi:10.3390/app122211523_

Round 1

Reviewer 1 Report

The paper entitled “3D Model Retrieval Algorithm Based on DSP-SIFT Descriptor and Codebook Combinationadopts Bag-of-Words into view-based 3D object retrieval and extracts DSP-SIFT from all images as objects feature. The manuscript is well organized and written and its contribution is almost enough to publish in Applied Sciences. However, some of the ideas those need to be thoroughly clarified:

1.    It is strongly recommended that the text be carefully checked and its problems be fixed.

2.    There are two paths of “codebook generation” and “quantization” in Figure 2, why?

3.    It is needed to explain the flowchart of Figure 2 in one paragraph.

4.    Determine the vertical axis of the Figure 5 and Figure 8.

5.    Explain the ETH-80 dataset in details in the section 4.1.

6.    It is strongly recommended to compare with newer methods.

7.    Some typos in the text:

a.     Page 3, Line 82, the sentence “After the codebook is generated, the …”, should be “After the codebook generation, the …”.

b.    Page 4, Line 120, in the sentence “After the feature extraction is completed, the approximate …”, it is better that this sentence be “After the feature extraction stage, the approximate …”. It is recommended to modify sentences like this sentence.

c.     Page 5, Line 143, insert a space between “Fx” and “represents”.

d.    Page 7, Line 194, I think “… Figure 6 …” should be “… Figure 7 …”.

e.    Page 8, Line 203, correct these sentences “Compare our method with the existing algorithm. Including MMGF, BoRW[24], BGM[27], AVC[28], CCFV[29], FDDL[30]. As shown in Figure 8, Figure 9.”

Author Response

We thank the associate editor and reviewers for their time and valuable comments. We are happy to know that all two reviewers agreed that the manuscript is well written. 

In terms of the reviewers’ concerns, we have addressed all of them. With the added materials, the current manuscript is better than the initial submission. We believe the quality is largely improved, thanks to the comments from the reviewers. The point-to-point responses are provided in the rest of this document

Reviewer 2 Report

The paper deals with 3d object retrieval which is a very hot research topic in the literature. In spite of this, the authors do not review related and previous works in a satisfactory depth. The authors cite very few recent papers related to 3d object retrieval. Here can be found a recent list of papers: https://dblp.org/search?q=3d+object+retrieval . However, the main problem with the manuscript is that the authors use only one very old benchmark database to present experimental results. This database was published in 2003. On the other hand, 3d object retrieval is a very rapidly evolving research field where researchers tend to use more recent benchmark database due to rapidly evolving nature of computer vision. Moreover, computer vision researchers work towards the usage of not outdated databases in general. Due to the very outdated benchmark database, it is impossible to the reviewer to judge whether the achieved results significant or interesting to the research community. Further, it can also occur that the proposed method do not address a specific gap in the field due to the outdated benchmark database. First of all, the authors should find a more recent benchmark database and add the experimental results measured on the chosen database to their experimental results section. Second, the authors should add a separate state-of-the-art review to the manuscript and contrast the proposed work to them.

Author Response

(The authors gave the same response as above.)

Round 2

Reviewer 1 Report

All comments were addressed by authors.

Reviewer 2 Report

I think the manuscript is suitable for publication now. The authors answered my questions and concerns.